# Differentially Private Estimation of Heterogeneous Causal Effects

**Fengshi Niu**                                                                FNIU@STANFORD.EDU
*Graduate School of Business*
*Stanford University*
*Stanford, CA 94305, USA*

**Harsha Nori**                                                               HANORI@MICROSOFT.COM
*Microsoft Research*
*Redmond, WA 98052, USA*

**Brian Quistorff** *                                                     BRIAN.QUISTORFF@BEA.COM
*U.S. Bureau of Economic Analysis*
*Washington, DC 20233, USA*

**Rich Caruana**                                                           RCARUANA@MICROSOFT.COM
*Microsoft Research*
*Redmond, WA 98052, USA*

**Donald Ngwe**                                                        DONALD.NGWE@MICROSOFT.COM
*Microsoft Research*
*Redmond, WA 98052, USA*

**Aadharsh Kannan**                                             AADHARSH.KANNAN@MICROSOFT.COM
*Microsoft Research*
*Redmond, WA 98052, USA*

**Editors:** Bernhard Schölkopf, Caroline Uhler and Kun Zhang

## Abstract

Estimating heterogeneous treatment effects in domains such as healthcare or social science often involves sensitive data where protecting privacy is important. We introduce a general meta-algorithm for estimating conditional average treatment effects (CATE) with differential privacy (DP) guarantees. Our meta-algorithm can work with simple, single-stage CATE estimators such as S-learner and more complex multi-stage estimators such as DR and R-learner. We perform a tight privacy analysis by taking advantage of sample splitting in our meta-algorithm and the parallel composition property of differential privacy. In this paper, we implement our approach using DP-EBMs as the base learner. DP-EBMs are interpretable, high-accuracy models with privacy guarantees, which allow us to directly observe the impact of DP noise on the learned causal model. Our experiments show that multi-stage CATE estimators incur larger accuracy loss than single-stage CATE or ATE estimators and that most of the accuracy loss from differential privacy is due to an increase in variance, not biased estimates of treatment effects.

**Keywords:** causal inference, differential privacy, interpretability, generalized additive models

---

*The views expressed in this paper are solely those of the authors and not necessarily those of the U.S. Bureau of Economic Analysis (BEA) or the U.S. Department of Commerce. Most of the work was completed at Microsoft Research before joining BEA.

## 1. Introduction

Understanding how treatment effects may vary across a population is critical for policy evaluations and optimizing treatment decisions. Estimating such treatment effect heterogeneity is common in medicine, marketing, and social science. Existing methods, however, may leak information about specific observations used in estimation. We propose a general way to estimate flexible heterogeneous treatment effect models in a privacy-preserving manner and show empirically how privacy guarantees impact estimation accuracy.

Using the potential outcomes framework (Neyman, 1923; Rubin, 1974), we focus on estimating the Conditional Average Treatment Effect (CATE) $\tau(x) = \mathbb{E}[Y(1) - Y(0)|X = x]$ where $Y$ is the outcome of interest, $X$ are observed covariates (features), $T$ is a binary treatment, and $Y(1), Y(0)$ are the potential outcomes under treatment and control. The setting can be experimental, where $T$ is a randomized manipulation, or observational, where $T$ is not explicitly randomized.

Privacy concerns arise when data includes sensitive information, such as healthcare data, finance data, or digital footprint. In these settings, mining population-wide patterns using algorithms without explicit privacy protection can reveal individuals' private information (Carlini et al., 2019; Melis et al., 2019; Shokri et al., 2017). The same concern extends to treatment effect estimation, especially CATE estimation, as the flexibly estimated CATE function at fine granularity could potentially reveal individual covariates, treatment status, or potential outcomes, all of which could be highly sensitive.

Differential privacy (DP) has emerged as a standard mathematical definition of privacy and has been widely adopted by U.S. Census Bureau and companies like Apple and Google for data publication and analysis (Erlingsson et al., 2014; Abowd, 2018). DP provides a robust, cryptographically motivated framework for tracking privacy loss and protects against privacy attacks even when an adversary has significant external information (Dwork et al., 2006). In this work, we show how to add DP guarantees to many popular CATE estimation frameworks.

The main contributions of this paper are:

1. We introduce a general meta-algorithm for multi-stage learning under privacy constraints. We apply this to CATE estimation, but it may be independently useful for other multi-stage estimation tasks, especially those using machine learning models to estimate nuisance parameters. Our meta-algorithm can work with simple, single-stage CATE estimators such as S-learner and more complex multi-stage estimators such as DR and R-learner.

2. We perform a tight privacy analysis by taking advantage of sample splitting in our meta-algorithm and the parallel composition property of differential privacy.

3. We study the trade-off between privacy and estimation accuracy using synthetic experiments and a differentially private interpretable ML model, DP-EBM (Nori et al., 2021), as the base learner. Our work clearly shows the change in estimation quality as privacy guarantees and sample size change. We show that flexible multi-stage CATE estimators incur larger accuracy loss than single-stage CATE or ATE estimators and that most of the accuracy loss from differential privacy is due to an increase in variance, not biased estimates of treatment effects.

### 1.1. Related Work

Our work builds on those from CATE estimation and differential privacy.

The modern CATE literature has focused on adapting flexible machine learning models to estimate CATE while utilizing sample-splitting to avoid biases due to overfitting. This includes methods that specify particular ML methods (Athey and Imbens, 2016; Wager and Athey, 2018)) and "meta-learners" that can utilize general ML algorithms (Nie and Wager, 2020; Künzel et al., 2019; Kennedy, 2020; Chernozhukov et al., 2018; Van der Laan and Rose, 2011). Our paper builds on the latter line of literature and provides a general recipe to make the existing meta-learners differentially private.

Differentially private algorithms have been proposed for many prediction tasks, such as classification, regression, and forecasting. These algorithms typically involve adding differential privacy guarantees to existing machine learning algorithms. For example, DP variants exist for linear models (Chaudhuri et al., 2011), decision tree classifiers (Jagannathan et al., 2009), stochastic gradient descent, and deep neural networks (Abadi et al., 2016). All of these models can be used as base components in our meta-algorithm.

Finally, there has been some initial work on differentially private causal inference methods. Lee et al. (2019) proposed a privacy-preserving inverse propensity score estimator for estimating average treatment effect (ATE). Their method ties with a specific differentially private prediction method for ATE estimation, which falls into the meta-algorithm framework in this paper. Komarova and Nekipelov (2020) studied the impact of differential privacy on the identification of statistical models and demonstrated identification of causal parameters failed in regression discontinuity design under differential privacy. Agarwal and Singh (2021) studied estimation of a finite-dimensional causal parameter under the local differential privacy framework. Our paper uses the central differential privacy framework, which gives weaker privacy protection and retains much higher data utility.

## 2. Background

### 2.1. Heterogeneous Treatment Effect Estimation

We assume our data consists of iid observations $Z_i = (Y_i, T_i, X_i)$, where $Y \in \mathbb{R}$ is the outcome, $T \in \{0, 1\}$ is a binary treatment, and $X \in \mathbb{R}^d$ are the observed covariates, and $i \in \{1, ..., n\}$. We define several nuisance functions:

$$e(x) = \mathbb{P}(T = 1 | X = x)$$
$$\mu(t, x) = \mathbb{E}[Y | T = t, X = x]$$
$$\eta(x) = \mathbb{E}[Y | X = x]$$

where $e(\cdot)$ is the propensity score, $\mu(\cdot, \cdot)$ is the joint response surface, and $\eta(\cdot)$ is the mean outcome regression. We aim to estimate $\tau(x) = \mu(1, x) - \mu(0, x)$, which is referred to as CATE. We will assume standard causal assumptions such as no unmeasured confounding ( $\{Y_i(0), Y_i(1)\} \perp\!\!\!\perp T_i | X_i$, i.e. conditional on $X$, treatment is independent of potential outcomes) and overlap of the treated and control propensity score distributions ($0 < e(x) < 1, \quad \forall x$). These are trivially satisfied for randomized experiments but typically takes domain knowledge to verify for observational studies.

### 2.2. Differential Privacy

Differential privacy is a widely adopted privacy definition based on the notion of a randomized algorithm's sensitivity to any single data point (Dwork et al., 2006). It is formally defined as:

**Definition 1 (($\varepsilon, \delta$)-Differential Privacy)** *A randomized algorithm* $\mathcal{A} : \mathbb{Z}^n \rightarrow \mathcal{H}$ *is ($\varepsilon$ ,$\delta$)-differentially private if for all neighboring datasets* $S, S' \in \mathbb{Z}^n$, *and for all* $\mathcal{O} \subset \mathcal{H}$:

$$\mathbb{P}(\mathcal{A}(S) \in \mathcal{O}) \leq e^{\varepsilon} \mathbb{P}(\mathcal{A}(S') \in \mathcal{O}) + \delta,$$

*where neighboring datasets are defined as two datasets that differ in at most one observation.*

Intuitively, differential privacy is a promise to each individual that the output of a DP algorithm will be approximately the same irrespective of its data. The privacy parameters $\varepsilon$ and $\delta$ parameterize the strength of the privacy guarantee – smaller values of $\varepsilon$ and $\delta$ ensure that the output distributions of $\mathcal{A}(S)$ and $\mathcal{A}(S')$ are closer, thereby ensuring more privacy. Algorithms typically achieve differential privacy guarantees by adding calibrated, symmetric noise to their outputs. Decreasing $\varepsilon$ or $\delta$ necessitates adding more noise to a model, which leads to an implicit trade-off between privacy and accuracy. We explore this trade-off for the CATE setting in Section 5.

We also leverage a refined notion of differential privacy called $f$-DP, which provides better mathematical tools for the privacy analysis of our algorithms (Dong et al., 2019). $f$-DP conceptualizes the privacy task as providing minimum error rates on statistical tests that would try to determine which of two potential samples were used to generate a statistic. It therefore parameterizes the privacy guarantees with a function, called a "trade-off function," that relates how Type-I and Type-II error can be traded-off in such a test. ($\varepsilon, \delta$)-DP is a special case of the $f$-DP framework. Our main theorem will be presented in terms of the more general $f$-DP space. Our experimental results will be shown in the more widely used ($\varepsilon, \delta$)-DP space. The following definitions are due to Dong et al. (2019), which we copy here for completeness.

**Definition 2 ($f$-Differential Privacy)** *Let $f$ be a trade-off function. A randomized algorithm $\mathcal{A}$ is said to be $f$-differentially private if for all neighboring datasets $S$ and $S'$,*

$$T\big(\mathcal{A}(S), \mathcal{A}(S')\big) \geq f.$$

**Definition 3 (Trade-off Functions)** *Let $P$ and $Q$ be any two probability distributions on the same space. Let $\phi$ be any (possibly randomized) rejection rule for testing $H_0 : P$ against $H_1 : Q$. Define the trade-off function $T(P, Q) : [0, 1] \rightarrow [0, 1]$ as*

$$\alpha \mapsto \inf_{\phi}\{1 - \mathbb{E}_Q[\phi] : \mathbb{E}_P[\phi] \leq \alpha\},$$

*where the infimum is taken over all rejection rules.*

## 3. Privacy-preserving CATE Learners

In this section, we present a meta-algorithm of conditional treatment effect estimation with simple and tight privacy guarantees. The meta-algorithm can be used with leading CATE learners in the literature, including DR-learner, R-learner, and S-learner, among others. Multiple sample splitting, i.e., using different parts of the sample to estimate different components of an estimator, is the key feature of this algorithm and enables the application of the parallel composition property of differential privacy. The privacy guarantee we give is learner-model-agnostic, i.e., it applies for any meta-learner in the literature and sub-algorithm modules as long as each sub-algorithm module is differentially private on its own.

## 3.1. Construction

In this subsection, we present our proposed meta-algorithm.

---

**Algorithm 1** A two-stage algorithm with sample splitting and data transformation

---

**input** Dataset $\mathcal{S} = \{Z_i\}_{i \in [n]} = \{(Y_i, T_i, X_i)\}_{i \in [n]}$, sample splitting ratio $\lambda_{1,1}, \ldots, \lambda_{1,k}$ and $\lambda_2$ with $\sum_{i=1}^{k} \lambda_{1,i} + \lambda_2 = 1$

**output** $\mathcal{M}(\mathcal{S}) = (\mathcal{A}_1(S_{1,1}), \ldots, \mathcal{A}_k(S_{1,k}), \mathcal{B} \circ \mathcal{T}(S_2; \mathcal{A}_1(S_{1,1}), \mathcal{A}_2(S_{1,2}), \ldots, \mathcal{A}_k(S_{1,k})))$, where $\mathcal{M}$ denotes the full algorithm, $\mathcal{A}_i, i \in [k]$ and $\mathcal{B}$ denote algorithm modules, $\mathcal{T}$ is a data transformation operator, $S_{1,1}, \ldots, S_{1,k}$ and $S_2$ form a partition of the full dataset with sample ratio $\lambda_{1,1}, \ldots, \lambda_{1,k}$ and $\lambda_2$

1: Sample splitting: let $(S_{1,1}, \ldots, S_{1,k}, S_2) = \texttt{Partition}(\mathcal{S}; \lambda_{1,1}, \ldots, \lambda_{1,k}, \lambda_2)$ be a partition of $\mathcal{S}$ that is chosen uniformly at random among all the partitions of $\mathcal{S}$ with sizes $n_{1,1} = \lambda_{1,1} \cdot n, \ldots, n_{1,k} = \lambda_{1,k} \cdot n$ and $n_2 = \lambda_2 \cdot n$
2: Run first-stage algorithms on disjoint sets of samples: $\mathcal{A}_1(S_{1,1}), \ldots, \mathcal{A}_k(S_{1,k})$
3: Construct data with transformed outcome, $\tilde{S}_2$, using $S_2$ and the output of first-stage algorithms: $\tilde{S}_2 = \{(X_i, \hat{\psi}_i)\}_{i \in S_2} = \mathcal{T}(S_2)$, in which $\hat{\psi}_i = \varphi(Z_i; \mathcal{A}_1(S_{1,1}), \mathcal{A}_2(S_{1,2}), \ldots, \mathcal{A}_k(S_{1,k}))$
4: Run the second-stage algorithm with the transformed outcome: $\mathcal{B}\left(\tilde{S}_2\right)$

---

Algorithm 1 is expressive enough to represent many leading CATE learners. We next show how to map the DR-learner, R-learner, and S-learner to this framework by specifying their $k, \mathcal{A}_1, \ldots, \mathcal{A}_k, \mathcal{T}$, and $\mathcal{B}$. Use $L(Y \sim X)$ to denote a regression estimator, which estimates $x \mapsto \mathbb{E}[Y|X = x]$.

- DR-learner from Kennedy (2020) uses a two-stage doubly robust learning approach. This learner divides the data into three samples $S_{1,1}, S_{1,2}, S_2$, so $k = 2$. The first stage estimates two nuisance functions. $\mathcal{A}_1$ uses $S_{1,1}$ to estimate the propensity score $\hat{e} = L_{1,1}(T \sim X)$. $\mathcal{A}_2$ uses $S_{1,2}$ to estimate the joint response surface $\hat{\mu} \sim L_{1,2}(Y \sim (T, X))$. The data transformation step uses $\hat{e}$ and $\hat{\mu}$ to construct the doubly robust score, $\hat{\psi}(Z) = \hat{\mu}(1, X) - \hat{\mu}(0, X) + \frac{Y - \hat{\mu}(1,X)}{\hat{e}(X)} \mathbb{1}(T = 1) - \frac{Y - \hat{\mu}(0,X)}{1 - \hat{e}(X)} \mathbb{1}(T = 0)$. The second stage, $\mathcal{B}$, uses the transformed data generated from $S_2$ to run the pseudo-outcome regression $\hat{\tau} = L_2(\hat{\psi} \sim X)$.

- R-learner from Nie and Wager (2020) uses a two-stage debiased machine learning approach. This learner divides the data into three samples $S_{1,1}, S_{1,2}, S_2$, so $k = 2$. The first stage estimates two nuisance functions. $\mathcal{A}_1$ uses $S_{1,1}$ to estimate the propensity score $\hat{e} = L_{1,1}(T \sim X)$. $\mathcal{A}_2$ uses $S_{1,2}$ to estimate the mean outcome regression $\hat{\eta} = L_{1,2}(Y \sim X)$. The data transformation step produces residuals from two regressions $\hat{\psi}(Z) = (Y - \hat{\eta}(X), T - \hat{e}(X))$. The second stage, $\mathcal{B}$, uses the transformed data generated from $S_2$ to run an empirical risk minimization $\hat{\tau} = \arg\min_\tau \sum_{i \in S_2} [\{Y_i - \hat{\eta}(X_i)\} - \{T_i - \hat{e}(X_i)\} \tau(X_i)]^2$.

- S-learner from Künzel et al. (2019) uses the whole dataset $S_2 = \mathcal{S}$ to estimate the joint response surface $\hat{\mu} \sim L_2(Y \sim (T, X))$ and then estimates $\hat{\tau}(x) = \hat{\mu}(1, x) - \hat{\mu}(0, x)$. In this case $k = 0$ and $S_2 = \mathcal{S}$.

**Multiple sample splitting** is a key design choice of Algorithm 1. The practice of using separate samples for estimating the nuisance functions in the first stage and the CATE function in the second stage is popular in the literature (Chernozhukov et al., 2018; Kennedy, 2020; Newey and Robins, 2018; Nie and Wager, 2020; Van der Laan and Rose, 2011). This is motivated by the fact that sample splitting avoids the bias due to overfitting. Kennedy (2020) and Newey and Robins (2018) point out that using different parts of the sample to estimate different components of an estimator, called double sample splitting, leads to simpler and sharper risk bound analysis. On top of these, we will see in Section 3.2 that multiple sample splitting leads to tight differential privacy guarantees by exploiting parallel composition and post-processing properties of differential privacy.

**Remark 4** Algorithm 1 releases all outputs of intermediates algorithm modules. For example, in the DR-learner case, it releases the estimated propensity score function, estimated regression function, and the final estimated CATE function to the public. From a practical point of view, releasing these intermediate outputs enables researchers to understand the data better and perform diagnostic tests. From a theoretical point of view, it is simpler to derive DP guarantees for compositions when intermediate outputs are released. Moreover, we cannot save any privacy budget by keeping the intermediate output private given the generality of our meta-algorithm. In this sense, it is privacy-budget-free to release all intermediates outputs.

**Remark 5** Many classical semiparametric estimators and popular approaches for applying machine learning methods to causal inference follow a multi-stage estimation scheme as Algorithm 1. Our Algorithm can be used to construct sample-split versions of these estimators. Notable examples include augmented inverse-propensity weighting (AIPW) estimators for average treatment effect (Robins et al., 1994), debiased machine learning for treatment and structural parameters (Chernozhukov et al., 2018; Foster and Syrgkanis, 2019), and estimating nonparametric instrumental variable models (Newey and Powell, 2003; Hartford et al., 2017) among others. Though the focus of this paper is specifically on conditional treatment effect estimation, our privacy guarantee applies to sample-split versions of all of these estimation algorithms.

### 3.2. Privacy Guarantees

In this subsection, we formally prove Algorithm 1 is private under the conditions that each of the sub algorithm modules is private. To give a tight analysis of the privacy guarantee, we adopt the $f$-DP framework proposed by Dong et al. (2019).

**Theorem 6** *Suppose*

1. *algorithm module $\mathcal{A}_i : \mathbb{Z}^{n_{1,i}} \to \mathcal{H}_{1,i}$ is $f_{1,i}$-DP for $i = 1, \ldots, k$, where $\mathcal{H}_{1,i}$ is the image space of $\mathcal{A}_i$,*

2. *algorithm module $\mathcal{B} : (\mathbb{X} \times \Psi)^{n_2} \to \mathcal{H}_2$ is $f_2$-DP, where $\Psi$ is the image space of the data transformation $\varphi : \mathbb{Z} \times \mathcal{H}_1 \times \ldots \times \mathcal{H}_k \to \Psi$,*

*then the composed meta-algorithm $\mathcal{M}$ in Algorithm 1 is $f$-DP with*

$$f = \min\{f_{1,1}, \ldots, f_{1,k}, f_2\}^{**},$$

*where $g^*(y) := \sup_{x \in \mathbb{R}} yx - g(x)$ denotes the convex conjugate of a generic function $g$ and $g^{**} = (g^*)^*$ denotes the double conjugate.*

A proof of Theorem 6 can be found in the appendix.

**Remark 7**  When all algorithm modules are $(\varepsilon, \delta)$-DP with the same $(\varepsilon, \delta)$, Theorem 6 suggests the whole algorithm is also $(\varepsilon, \delta)$-DP.

**Remark 8**  Multiple sample splitting is essential for this simple yet tight analysis of differential privacy. More precisely, it enables the usage of the parallel composition property of differential privacy and saves the privacy budget. Without sample splitting, we will need to pay additional sequential composition cost in the differential privacy analysis. For example, if composed naively without sample splitting, the privacy loss of the corresponding DR-learner with all modules satisfying $(\varepsilon, \delta)$-DP would be $3 \cdot \varepsilon$ instead of $\varepsilon$.

**Remark 9**  The privacy guarantee of Algorithm 1 provided by Theorem 6 is agnostic about what DP algorithm modules one uses. For example, one could use DP deep neural networks (trained using DP stochastic gradient descent) as the base learner to construct DP CATE learners. In the following sections, we focus on DP CATE learners with a specific interpretable DP prediction algorithm as the base learner and conduct experiments to study their empirical performance.

## 4. DP-EBM CATE Learners

In this section, we introduce DP-EBMs as the base learner in the meta-algorithm and present the full algorithm of a specific differentially private CATE leaner we call DP-EBM-DR-learner.

### 4.1. DP-EBM

DP-EBM is a differentially private and interpretable machine learning algorithm recently introduced by Nori et al. (2021), which adds differential privacy guarantees to Explainable Boosting Machines (EBM) (Nori et al., 2019; Lou et al., 2012, 2013). EBMs and DP-EBMs belong to the family of Generalized Additive Models (GAMs), which are restricted models of the form:

$$g(\eta(x)) = \alpha + f_1(x_1) + ... + f_d(x_d),$$

where $\eta(x) = \mathbb{E}[Y|X = x]$ is the regression function, $f_i$ is a univariate function that operates on each input feature $x_i$, $\alpha$ is an intercept, and $g$ is a link function that provides the relationship between the additively separable function to the regression function and in doing so adapts the model to different settings like classification and regression. GAMs are a relaxation of generalized linear models in which each function $f_i$ is restricted to linear. In the GAM setting, the models are allowed to flexibly learn complex functions of each feature, which has been shown to yield more expressive and accurate models (Chang et al., 2021; Caruana et al., 2015; Nori et al., 2019). Moreover, the additive structure of the models and inability to learn complex interactions between features (e.g., $f(x_1, x_2, x_3)$) allows GAMs to remain interpretable. At prediction time, the contribution of each feature $i$ is exactly $f_i(x_i)$. These term contributions can be directly compared, sorted, and reasoned about. In addition, each function $f_i$ can be visualized as a graph to show how the model's predictions change with varying inputs.

DP-EBMs add privacy guarantees to EBMs by adding calibrated Gaussian noise during each iteration of the training process. This noise has been shown to cause distortion in the learned individual feature functions $\hat{f}_i$, especially under strong privacy guarantees.

We use DP-EBMs to highlight our DP CATE learning algorithm for two main reasons: DP-EBMs have been shown to yield high accuracy under strong privacy guarantees on tabular datasets, and the interpretability of DP-EBMs is useful to showcase how varying privacy guarantees affect CATE estimation (Nori et al., 2021).

## 4.2. DP-EBM-DR-learner

Previously, Algorithm 1 outlined a general recipe to construct DP CATE learners. Now that we have introduced DP-EBM, a concrete prediction algorithm with privacy guarantee, we will present a complete CATE estimator that we call DP-EBM-DR-learner.

---

**Algorithm 2** DP-EBM-DR-learner

**input** Dataset $\mathcal{S} = \{Z_i\}_{i\in[n]} = \{(Y_i, T_i, X_i)\}_{i\in[n]}$, privacy parameters $\varepsilon \in (0,\infty)$, $\delta \in (0,1)$, sample splitting ratio $\lambda_{1,1}, \ldots, \lambda_{1,k}$ and $\lambda_2$ with $\sum_{i=1}^{k} \lambda_{1,i} + \lambda_2 = 1$,

**output** $\mathcal{M}(\mathcal{S}) = (\hat{e}, \hat{\mu}, \hat{\tau})$, where $\mathcal{M}$ denotes the full algorithm

1: Sample splitting: let $(S_{1,1}, S_{1,2}, S_2) = \texttt{Partition}(\mathcal{S}; \lambda_{1,1}, \lambda_{1,2}, \lambda_2)$ be a partition of $\mathcal{S}$ that is chosen uniformly at random among all the partitions of $\mathcal{S}$ with sizes $n_{1,1} = \lambda_{1,1} \cdot n, n_{1,2} = \lambda_{1,2} \cdot n$ and $n_2 = \lambda_2 \cdot n$

2: Run first-stage algorithms on disjoint sets of samples:

    (a) Use $S_{1,1}$ and DP-EBM classifier to estimate propensity scores $\hat{e} = \text{DP-EBM}(T \sim X; S_{1,1}, \varepsilon, \delta)$

    (b) Use $S_{1,2}$ and DP-EBM regression to estimate joint response surface $\hat{\mu} = \text{DP-EBM}(Y \sim (T, X); S_{1,2}, \varepsilon, \delta)$

3: Construct data with transformed outcome, $\tilde{S}_2$, using $S_2$ and the output of first-stage algorithms: $\tilde{S}_2 = \{(X_i, \hat{\psi}_i)\}_{i\in S_2}$, in which $\hat{\psi}_i = \hat{\mu}(1, X_i) - \hat{\mu}(0, X_i) + \frac{Y_i - \hat{\mu}(1, X_i)}{\hat{e}(X_i)}\mathbb{1}(T_i = 1) - \frac{Y_i - \hat{\mu}(0, X_i)}{1 - \hat{e}(X_i)}\mathbb{1}(T_i = 0)$

4: Use the transformed data $\tilde{S}_2$ and DP-EBM regression to estimate CATE $\hat{\tau} = \text{DP-EBM}(\hat{\psi} \sim X; \tilde{S}_2, \varepsilon, \delta)$

---

DP-EBM-DR-learner is $(\varepsilon, \delta)$-DP by theorem 6 and remark 7.

Similarly, we can construct DP-EBM-R-learner and DP-EBM-S-learner following Section 3.1 and use DP-EBM for fitting the regression model (or empirical risk minimization).

Our main focus here is DP-EBM-DR-learner due to the favorable theoretical properties of DR-learner, including double robustness (Kennedy, 2020), connection to semiparametric efficiency (Robins et al., 1994), and its interpretability. More precisely, DR-learner will always estimate the projection of the true CATE function onto the space of functions over which we optimize in the final regression, even if the true CATE function does not lie in this function space. This fact partially mitigates the problem of model misspecification. We pick DP-EBM prediction algorithm due to its exact model interpretability and high accuracy under privacy constraints.

R-learner shares many of the theoretical properties of DR-learner. DP-EBM-R-learner also has similar empirical performance as DP-EBM-DR-learner in the following experiments.

We treat S-learner as the baseline due to its simplicity. Since the approach is relatively simple and does not use estimated propensity scores, S-learner often has larger bias (integrated squared bias) and smaller variance (integrated variance) for moderate sample sizes. DP-EBM-S-learner reduces to a differentially private average treatment effect (ATE) estimator. This happens because the functional form restriction of DP-EBM enforces the CATE estimate of S-learner to be a constant $\hat{\tau}(x) = \hat{\mu}(1, x) - \hat{\mu}(0, x) = \hat{f}_T(1) - \hat{f}_T(0)$, where $\hat{f}_T$ is the estimated univariate function of the treatment. As a result, DP-EBM-S-learner, an ATE estimator, has a large irreducible bias and a small variance for estimating CATE.

## 5. Experiments

In this section, we present experimental results for DP CATE learners using data from a real-world experiment with synthetic treatment effect in addition to five simulated datasets. Imposing privacy constraints necessarily compromises accuracy in estimating CATE. Understanding this privacy-accuracy trade-off is the main goal of our experiments.

We present our analysis to illustrate how DP noise affects the estimated CATE function. We show how different DP CATE learners applied to different sample sizes are affected differently by varying the level of privacy constraints. Further, we elaborate on the privacy-accuracy trade-off by decomposing the increase in mean squared error (MSE) into increases in bias and variance. The code for all experiments in this section is publicly available at https://github.com/FengshiNiu/DP-CATE.

### 5.1. Setups

#### 5.1.1. ALGORITHM SPECIFICATION

In our experiments, we evaluate three DP CATE learners: DP-EBM-DR-learner, DP-EBM-R-learner, and DP-EBM-S-learner. For the first two, the sample splitting ratio is set to $(\lambda_{1,1}, \lambda_{1,2}, \lambda_2) = (0.25, 0.25, 0.5)$, which means estimating propensity score, outcome regression, and CATE function each uses a quarter, a quarter, and a half of the whole sample, respectively. We use the DP-EBM classifier and regressor implemented in the python package `interpret`[1] introduced in Nori et al. (2019) with all hyperparameters set to their default values. Except for propensity scores, which are estimated using a DP-EBM classifier, all other functions in the experiments are fitted using a DP-EBM regressor.

#### 5.1.2. DATA DESCRIPTION

We describe how we generate different datasets $\{(Y_i, T_i, X_i)\}_{i \in \mathcal{S}}$ in all experiments using the following language. We use $P_X$ to denote the covariate distribution, $e(x) := P(T = 1 | X = x)$ to denote the propensity score, and $b(x) := \mathbb{E}[Y(0) | X = x]$ for the control baseline function.

The main dataset we consider is a real dataset from Arceneaux et al. (2006), which studies the effect of paid get-out-the-vote calls on voter turnout. We refer to this dataset as the voting data. This data is originally generated from a stratified experiment with varying treatment probability and has previously been used for comparing and evaluating the performance of different causal effect estimators (Arceneaux et al., 2006; Nie and Wager, 2020). Nie and Wager (2020) points out that the

---

[1] https://github.com/interpretml/interpret

true treatment effect in this data is close to non-existent, so we can spike the original dataset with a synthetic treatment effect $\tau(\cdot)$ to make the task of estimating heterogeneous treatment effects non-trivial. The modified dataset therefore has the covariate distribution and propensity function from the true data and has a synthetic treatment effect. Because we know the true synthetic treatment effect $\tau(\cdot)$, we can compare the estimated $\hat{\tau}(\cdot)$ to it and evaluate its performance.

Following Nie and Wager (2020), we focus on a subset of 148,160 samples containing 59,264 treated units and 88,896 controls. The covariates have 11 variables, which include state, county, age, gender, past voting record, etc. The synthetic treatment effect is set to $\tau(X_i) = -\frac{\text{VOTE00}_i}{2+100/\text{AGE}_i}$, where VOTE00i indicates whether the $i$-th unit voted in the year 2000. The synthetic treatment effect is added by strategically flipping the binary outcome. Denote the original outcome in the data by $Y^*$. With probability $1 - \tau(X_i)$, we set $Y_i(0) = Y_i(1) = Y^*$. With probability $\tau(X_i)$, we set $Y_i(0) = 1, Y_i(1) = 0$. Finally, we set $Y_i = Y_i(T_i)$. The treatment heterogeneity as measured by $\text{Var}(\tau(X)) = 0.016$ is moderate. This number can also be interpreted as the MSE of using the true ATE as a CATE estimator.

All five other datasets are generated by simulation based on the same recipe:

$$X_i \sim P_X, \ T_i | X_i \sim \text{Ber}(e(X_i)), \epsilon_i \sim \text{N}(0, 1)$$
$$Y_i = b(X_i) + T_i \cdot \tau(X_i) + \epsilon_i.$$

We specify setups $A$ to $E$ with different $(P_X, b, e, \tau)$ to cover a variety of settings.

| Setup | $P_X$ | $b$ | $e$ | $\tau$ |
|---|---|---|---|---|
| A | $\text{U}(0,1)^d$ | $\sin(\pi x_1 x_2) + 2(x_3 - 0.5)^2$ $+ x_4 + 0.5x_5$ | $\text{trim}_{0.1}\{\sin(\pi x_1 x_2)\}$ | $(x_1 + x_2)/2$ |
| B | $\text{N}(0, I_d)$ | $\max\{x_1 + x_2, x_3, 0\}$ $+ \max\{x_4 + x_5, 0\}$ | $0.5$ | $x_1 + \log(1 + e^{x_2})$ |
| C | $\text{N}(0, I_d)$ | $2\log(1 + e^{x_1 + x_2 + x_3})$ | $1/(1 + e^{x_2 + x_3})$ | $1$ |
| D | $\text{N}(0, I_d)$ | $\max\{x_1 + x_2 + x_3, 0\}$ $+ \max\{x_4 + x_5, 0\}$ | $1/(1 + e^{-x_1} + e^{-x_2})$ | $\max\{x_1 + x_2 + x_3, 0\}$ $- \max\{x_4 + x_5, 0\}$ |
| E | $\text{N}(0, \Sigma_d)$ | $\sum_{i=1}^{6} i x_i + x_1 x_6$ $+ \mathbb{1}(-0.5 < x_3 < 0.5)$ | $1/(1 + e^{x_1 + x_6})$ | $1/(1 + e^{x_1}) - x_2$ $+ \sum_{i=3}^{6} x_i$ |

Table 1: Experiment Setup

For all setups, $d = 6$. The trim function is specified as $\text{trim}_{0.1}(x) = \min\{\max\{0.1, x\}, 0.9\}$. Setup A, B, C, and D are from Section 6 of Nie and Wager (2020). Setup A has a complicated baseline function. Setup B is a randomized controlled trial with $e = 0.5$. Setup C has a constant treatment effect with $\tau = 1$. Setup D has a nondifferentiable treatment effect function. Setup E has correlated covariates with $\Sigma_d$ generated from a random data generator. Its baseline function is discontinuous and contains an interaction term between variables.

### 5.1.3. EXPERIMENT HYPERPARAMETERS

We run all three DP CATE learners on all six datasets with seven levels of training sample size growing exponentially $\{500, 1000, 2000, 4000, 8000, 16000, 32000\}$. For setups A-E, we generate a fixed test sample of size 250000 using their data generating process and use it throughout the

experiments. For the voting data, we construct the training data by random sampling (stratified by treatment) from the whole data and use all data left as the test set. The privacy parameter $\delta$ is fixed as $10^{-5}$ and $\varepsilon$ varies between $\{1, 2, 4, 8, 16\}$. $\varepsilon = 16$ is seen as approximately non-private since the privacy guarantee is extremely weak. $\varepsilon = 1$ and $\varepsilon = 2$ are seen as strong guarantees.

For each specification, we train the algorithm twice on two separate training sets with the specified sample size. We then generate three empirical MSE corresponding to the predicted CATE using each of the two trained algorithms separately and the average of these two algorithms on the test set. These three MSE are denoted as $\widehat{\text{MSE}}_1, \widehat{\text{MSE}}_2, \widehat{\text{MSE}}_{\text{avg}}$. We generate estimated $\widehat{\text{MSE}} = (\widehat{\text{MSE}}_1 + \widehat{\text{MSE}}_2)/2$, $\widehat{\text{Bias}} = 2\widehat{\text{MSE}}_{\text{avg}} - \widehat{\text{MSE}}$, $\widehat{\text{Var}} = 2\widehat{\text{MSE}} - \widehat{\text{Bias}}$. Here $\widehat{\text{Bias}}$ is meant to estimate the integrated squared bias $\int (\mathbb{E}_{\mathcal{S}}[\hat{\tau}_{\mathcal{S}}(x)] - \tau(x))^2 dF(x)$, in which $\mathbb{E}_{\mathcal{S}}[\hat{\tau}_{\mathcal{S}}(x)]$ denotes the expected estimate at point $x$ over the distribution of data $\mathcal{S}$. $\widehat{\text{Var}}$ is meant to estimate the integrated variance $\int \text{Var}(\hat{\tau}_{\mathcal{S}}(x))dF(x)$. This calculation is motivated by the fact that MSE of the average prediction is the sum of the integrated squared bias plus half of the integrated variance. For each of the specification we run this process five times and report the average $\widehat{\text{MSE}}, \widehat{\text{Bias}}, \widehat{\text{Var}}$.

## 5.2. Insights from Experiment Results

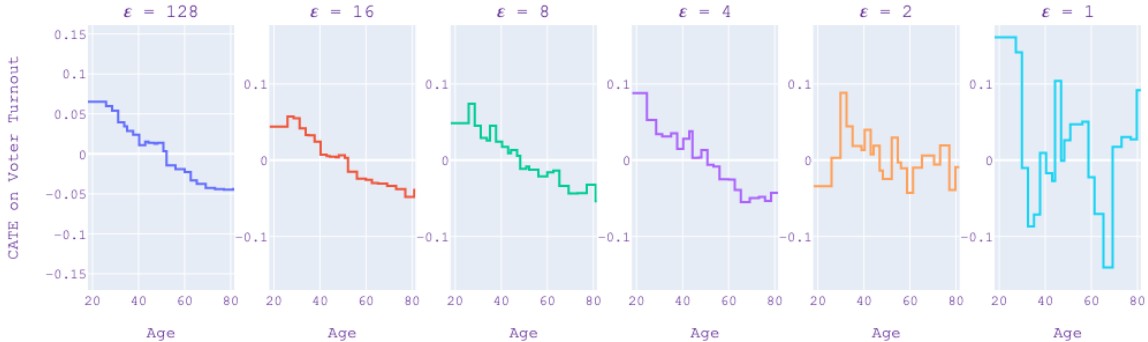

Figure 1: Estimate of CATE on voter turnout as a function of Age (one of the 11 features) using voting data with sample size 16000 and DP-EBM-DR-learner with varying $\varepsilon$.

Figure 1 shows the shape function of the conditioning variable "Age" that the DP-EBM-DR-learner learned from the voting data with sample size 16000 at five different levels of $\varepsilon$. The main purpose of this figure is to transparently show the effect of adding differential privacy constraints on the estimated CATE function. Since the true shape function corresponds to the projection of the synthetic CATE function $\tau^*(X_i) = -\frac{\text{VOTE00}_i}{2 + (100/\text{AGE}_i)}$ onto the additive separable function space, it is smooth and monotonically decreasing. DP-EBM-DR-learner learns this well when the algorithm is nonprivate or the privacy is low with $\varepsilon = 128$, 16, or 8. When privacy gets stronger with $\varepsilon = 4$ or 2 or 1, the estimated shape functions become jumpier and less smooth. Intuitively, this happens because differential privacy is achieved by adding Gaussian noise in the training process. Higher levels of noise under strong privacy lead to less well-behaved nonparametric function estimates. Similar behavior has been observed when DP-EBMs are applied to general prediction tasks (Nori et al., 2021).

Figure 2 plots the MSE of DP-EBM-DR-learner, DP-EBM-R-learner, and DP-EBM-S-learner under seven different sample sizes of the voting data at five different levels of privacy. Each curve

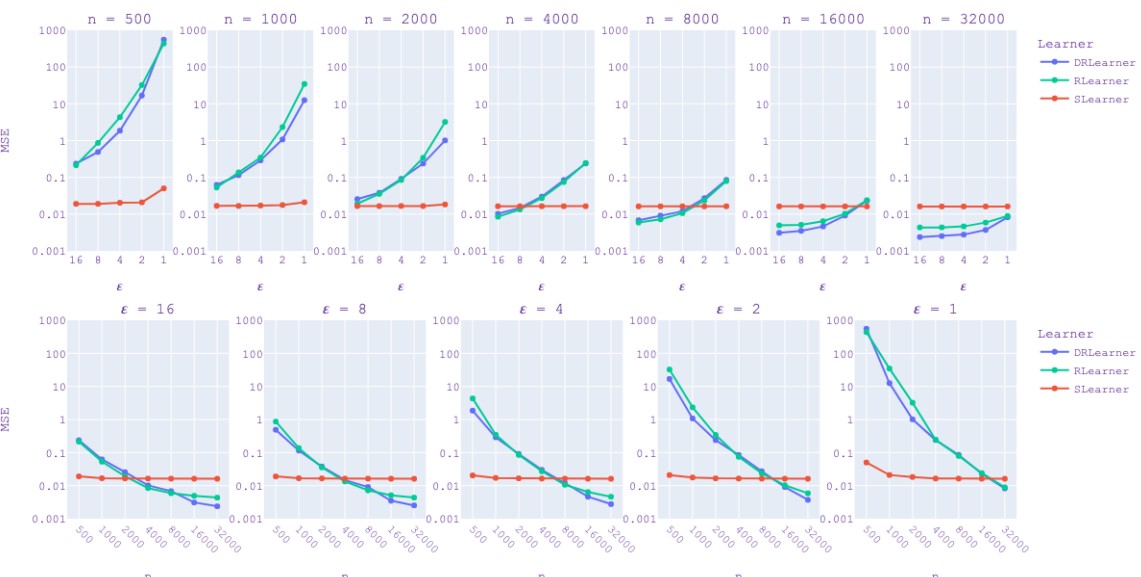

Figure 2: Comparison of DP-EBM-DR-learner, DP-EBM-R-learner, and DP-EBM-S-learner on voting data with varying DP guarantee $\varepsilon$ and sample size $n$. All MSE values are averages of 25 runs reported on an independent test set.

in the upper row shows how MSE increases as the privacy gets stronger at a given sample size. Each curve in the bottom row shows how MSE decreases as the sample size increases at a given privacy level.

The performance of the less flexible DP-EBM-S-learner stays remarkably stable with MSE equal to about 0.016 for most designs and is larger than this only at only $n = 500$ and $\varepsilon = 1$ where the sample size is the smallest and the privacy is the strongest. This happens because DP-EBM-S-learner outputs a constant scalar estimate of ATE, which is easily estimable with hundreds of samples despite the privacy constraint. In fact, its MSE approximately equals $\text{Var}(\tau^*(X)) = 0.016$, which is the MSE of the true ATE.

The more flexible DP-EBM-DR-learner and DP-EBM-R-learner are much more sensitive to the privacy parameter $\varepsilon$. This is especially true when the sample size $n$ is small. MSE increases by a factor of $10^3$ going from $\varepsilon = 16$ to $\varepsilon = 1$ at $n = 500$; while it only increases by a factor less than 10 from $\varepsilon = 16$ to $\varepsilon = 1$ at $n = 32000$. For the intermediate sample size $n = 16000$, DR-learner and R-learner are better until privacy is at its strongest at $\varepsilon = 1$, as also suggested by Figure 1.

A researcher interested in exploiting heterogeneity in treatment effects under privacy constraints will need to decide whether to use a flexible CATE estimator or a simple ATE estimator. The lower plot shows that when there is almost no privacy at $\varepsilon = 16$, flexible CATE estimators perform better with 3000 or more samples on this dataset. As the privacy requirements get stronger, it takes more samples for the flexible CATE estimator to perform better. At $\varepsilon = 1$ the simple ATE estimator outperforms the CATE estimators until about 20000 samples on this dataset.

Figure 3 shows the mean squared error, integrated squared bias, and integrated variance values of the DP-EBM-DR learner on all six datasets. Moving from no privacy to strong privacy, the variance of this learner often increases by about a factor of 10 while the bias changes by at most a factor of 2. This bias-variance decomposition reveals that accuracy loss due to privacy constraints

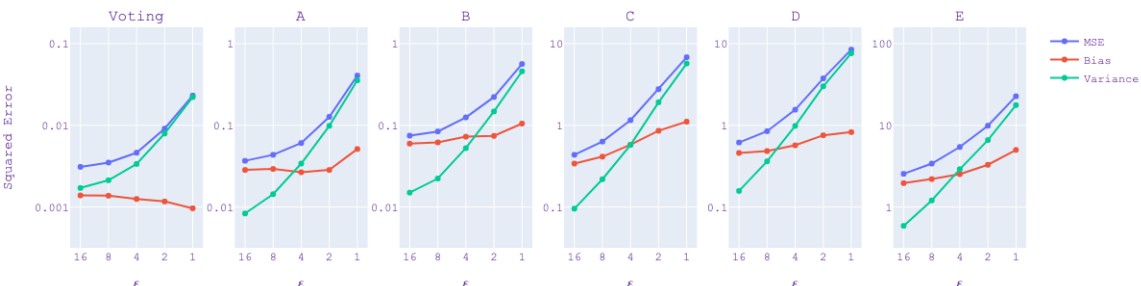

Figure 3: Performance of DP-EBM-DR-learner on six datasets. Datasets Voting, A, and B use 16000 samples, and C, D, and E use 4000 samples. All mean squared error (MSE), bias, and variance values are averages of 25 runs from an independent test set.

comes mostly from the increase in variance. In other words, the privacy-variance trade-off captures most of the privacy-accuracy trade-off. This finding is in favor of CATE estimation under privacy constraints, because bias is a stronger concern than variance in estimation of causal effects which often deals with critical policy, economic or healthcare issues.

Our main experimental findings can be summarized as:

1. Stronger privacy constraints lead to noisier and jumpier estimated CATE functions.

2. Most of the additional accuracy loss due to privacy can be attributed to a significant increase in variance, while bias often increases at a much slower rate.

3. Flexible, multi-stage CATE learners incur larger accuracy loss from differential privacy than less flexible, single-stage CATE or ATE learners. It takes a significantly larger sample size to exploit treatment effect heterogeneity under strong privacy constraints.

## 6. Discussion and Conclusion

In this paper, we presented a meta-algorithm for differentially private estimation of the Conditional Average Treatment Effect (CATE). We showed that the meta-algorithm worked with a variety of CATE estimation methods and provided tight privacy guarantees. We pointed out multiple sample splitting, which enabled the usage of the parallel composition property of differential privacy and saved privacy budget, as the key feature of the meta-algorithm. We implemented a fully specified CATE estimator using DP-EBMs as the base learner. DP-EBMs were interpretable, high-accuracy models with privacy guarantees, which allowed us to directly observe the impact of DP noise on the learned causal model. We conducted experiments that varied the size of the training data and the strength of the privacy guarantee to study the trade-off between privacy and CATE estimation accuracy. Our experiments indicated that most of the accuracy loss from differential privacy in the high-privacy, low-to-modest sample size regime was due to an increase in variance rather than bias. The experiments also showed that flexible, multi-stage CATE learners incurred larger accuracy loss from differential privacy than less flexible, single-stage CATE or ATE learners. It would be interesting to complement these empirical findings by characterizing theoretically the trade-off between privacy and accuracy for estimating CATE.

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
