# OpenReview forum: "Differentially Private Estimation of Heterogeneous Causal Effects"
_cclear.cc/CLeaR/2022/Conference — CLeaR 2022 Oral_

### Official Review · Reviewer_Qg9u · 2021-11-20

**Confidence:** 4
**Overall Score:** 6

**Main Review:**

Strengths:

1. Conditional, or heterogeneous, average treatment effect estimation (CATE) is a natural problem that differential privacy is of great concern, given the essential role of CATE in individualized decision making. Therefore, this paper is indeed trying to make progress on an important problem. The topic is very fresh -- there are indeed only a few papers that consider differentially private causal inference.

2. The section before numerical experiments is relatively well-written, not dense with technical jargon, and accessible to people working on either causal inference or privacy data analysis.

Weaknesses:

1. The insight offered by the only theorem proved (Theorem 6) is quite limited. It is good to calculate the "tight" DP indices for the entire estimation procedure, but the only concrete example in the paper is about the trivial scenario: if all sub-routines are $(\epsilon, \delta)$-DP, then the entire procedure is $(\epsilon, \delta)$-DP. The motivation for proving this theorem is not strong, in my opinion.

2. The numerical experiment section lacks sufficient detail. The authors did not provide any details on the DP-nuisance parameter estimation algorithms. It is difficult to judge the value of the experiments without knowing these details.

3. The potential research questions suggested in this paper are only based on one simulation study. Since this paper is not about proving important theoretical results, more comprehensive simulation experiments should be performed. A pure "intuition + simulation studies" based paper, even without any theoretical guarantees, can still be very valuable to the community. But the experiments must be very comprehensive and aim at pushing the boundaries of the data generating mechanisms as hard as one could.

4. The insights offered by the experiments are not very pedagogical. When the sample size is large, complicated methods can perform better than simpler methods; when the sample size is low, simpler methods might be more stable so one might be willing to sacrifice some bias. Isn't it true for almost any statistical practice? How much more could we learn by following these directions, except that we add the DP component?

5. The type of differential privacy that the authors tackled is black-box. For instance, probably a most natural DP-related setting is what if some of the key confounders have to be taken to be differentially private while modeling the nuisance parameters and the final CATE function. A more concrete example will be more useful to illustrate the significance of either the theoretical or the empirical contribution of this paper.

Overall, the paper is trying to tackle an interesting problem. Unfortunately, the current "signal" in the paper is quite low: the only theorem is not a core contribution to the field; the simulation experiments are not solid or comprehensive. As a result, I would not recommend acceptance of this paper.


**Summary:**

This paper studied the indices of DP for the entire procedure of estimating CATE when each component of the procedure is made DP and suggested some interesting directions based on results of numerical experiments.

---

> ### Author Response · Authors · 2021-12-03
> **Reply to Reviewer Qg9u Part 2**
>
> 4. *"The insights offered by the experiments are not very pedagogical..."*
>
> Our experiments deliver three takeaways as summarized on page 12.
> The reviewer seems to miss the first two takeaways, both of which are important for characterizing the effect of differential privacy on CATE estimation. The reviewer only comments on the third takeaway and unfortunately misinterprets it.
>
> We restate the first two takeaways here:
>
> (1) As expected, stronger privacy constraints lead to noisier and jumpier estimated CATE functions.
>
> (2) Most of the additional accuracy loss due to privacy can be attributed to a significant increase in variance, while bias increases at a much slower rate. This is important because bias is a much stronger concern than variance --- in CATE estimation which often deals with critical policy, economic or healthcare issues, it is much worse to confidently believe a wrong thing than to have too much uncertainty.
>
> Our third takeaway in the paper is not saying more complicated and data-hungry methods can only outperform simple methods when the sample size is large.
> Instead, our third takeaway in the paper is about how differential privacy, the new element we introduce to the CATE estimation problem, changes the tradeoff between more complicated vs simpler CATE estimation methods.
> We restate the third takeaway in the paper here ``Flexible, multi-stage CATE learners incur larger accuracy loss from differential privacy than less flexible, single-stage CATE or ATE learners.
> It takes a significantly larger sample size to exploit treatment effect heterogeneity under strong privacy constraints''.
> We consider this takeaway instructive for shaping researchers' intuition on how differential privacy affects a spectrum of CATE estimation procedures differently.
> Experiment results behind this takeaway are presented in Figure 2 on Page 11. The text following Figure 2 presents the discussion with more detail.
>
> 5. *"The type of differential privacy that the authors tackled is black-box..."*
>
> The $(\varepsilon, \delta)$-DP and $f$-DP definitions of differential privacy used in this paper are standard and have been discussed extensively in the literature.
> This paper states both definitions formally in definitions 1, 2, and 3.
> These definitions are meant to quantify the worst-case privacy loss of data of each individual, which includes its covariates, treatment status, and potential outcomes.
> The mathematical definition and the interpretation of differential privacy this paper considers are both transparent and well studied.
>
> *"A most natural DP-related setting is what if some of the key confounders have to be taken to be differentially private..."*
>
> The reviewer suggests an alternative definition of privacy, which could be interesting, but alternate definitions of DP fall well outside the scope of this paper.
> More specifically, the reviewer mentions a notion of confounder-specific privacy.
> We are not aware of any definition addressing confounder-specific privacy in the differentially privacy literature.
> It is not clear to us against what kind of privacy attacks this notion of privacy is robust.
> In case this newly proposed notion of privacy makes sense, the DP guarantee of the CATE estimation algorithms in our paper will naturally be an upper bound of the kind of privacy loss the reviewer is concerned about.
>
> *"A more concrete example will be more useful..."*
>
> We completely agree that a more concrete example that illustrates the theoretical and empirical significance of the work would be useful and we will add such an example to the final draft.  Thank you for suggesting this.

---

> > ### Comment · Reviewer_Qg9u · 2021-12-26
> > **Thank you for your rebuttal**
> >
> > I want to thank the authors for the comprehensive rebuttal to my concerns. The rebuttals are convincing. Now I do agree now with the authors and other referees that this paper addresses quite important and novel questions in causal inference. As a result, I'll increase my evaluation to "marginally above the acceptance threshold".

---

> ### Author Response · Authors · 2021-12-03
> **Reply to Reviewer Qg9u Part One**
>
> 1. *"The insight offered by Theorem 6 is quite limited..."*
>
> We believe the motivation for Theorem 6 is strong and the insight it offers is nontrivial. The paper is about CATE estimation under differential privacy guarantees, which requires that any algorithm be accompanied by a formal proof of privacy. This is what Theorem 6 is all about.
>
> Theorem 6 is nontrivial. In particular, ``if all sub-routines are $(\varepsilon, \delta)$-DP, then the entire procedure is $(\varepsilon, \delta)$-DP'' mentioned by the reviewer, is not true for arbitrary multi-stage algorithms.
> One of the most important contributions of this paper is pointing out that multiple sample splitting, a key feature of our CATE  estimation procedure, is essential for a simple yet tight analysis of differential privacy.
> More precisely, multiple sample splitting enables the usage of the parallel composition theorem introduced in Smith et al. (2021).
> Without sample splitting, we will need to pay additional sequential composition cost in the differential privacy analysis. For example, if composed naively without sample splitting, the privacy loss of the corresponding DR-learner would be $3 * \varepsilon$ instead of $1 * \varepsilon$. **The simplicity of the DP guarantee proved in Theorem 6 is the result of careful algorithm design together with the utilization of state-of-the-art DP analytical tools, not because the algorithm and analysis is trivial.** We will edit the paper to make this more clear.
>
> Finally, we prove the DP guarantee in the f-DP framework and then specialize to $(\varepsilon, \delta)$-DP framework mainly because f-DP framework enables a tighter analysis than $(\varepsilon, \delta)$-DP.
> Since f-DP and the related Gaussian differential privacy framework covers a variety of DP frameworks like, e.g. $(\varepsilon, \delta)$-DP and Renyi-DP, and have recently gained increasing popularity in the differential privacy community, the way we present the DP analysis is also meant to be friendly to researchers used to various DP analytical tools.
>
> 2. *"The numerical experiment section lacks sufficient detail..."*
>
> Details of our experimental setup, including choices of hyperparameters, dataset specifications, sample sizes, values of $\varepsilon$, and metrics tracked, are all reported in Sections 5.1.1, 5.1.2, and 5.1.3.
>
> *"The authors did not provide any details on the DP-nuisance parameter estimation algorithms..."*
>
> The details of the machine learning method used for all stages of the CATE estimator -- DP-EBM -- are introduced in Section 4.1 of our paper. Algorithm 2 step 2 explicitly specifies the method we use for DP-nuisance parameter estimation with DP-EBM.
>
> 3. *"The potential research questions suggested in this paper are only based on one simulation study..."*
>
> We cannot agree with the reviewer's claim. Section 5 of our paper presents experiments of the proposed algorithms on six different DGPs.
> One of the six datasets is a real dataset from Arceneaux et al. (2006), which studies the effect of paid get-out-the-vote calls on voter turnout.
> The other five datasets are simulated datasets.
> Overall, the six datasets cover the DGPs with binary outcomes or continuous outcomes, independent or correlated covariates distribution, simple or complicated baseline control outcome functions, randomized control trials or complicated treatment assignment regimes, and smooth or nonsmooth CATE functions.
> More detail can be found in table 1 and the paragraph at the bottom of page 9 of the paper.
> In this sense, our simulation study challenges the CATE estimation algorithm in many different aspects.
> For each of the datasets, we also consider seven different sample sizes.
>
> Unlike evaluation of a prediction algorithm, evaluation of CATE estimation procedure often requires the knowledge of the ground truth CATE function (Caveat: There is early literature on causal evaluation, but results there are still at an early stage.).
> This intrinsic challenge implies that researchers will need to use simulated datasets or semi-synthetic datasets in which they know the true CATE functions for evaluation.
> Given this situation, we collect datasets and DGPs used in recent papers Nie and Wager (2021) and Kennedy (2020) in the CATE literature for our data experiments.
> Our experiments are on par with the state-of-the-art in the CATE literature.

---

### Official Review · Reviewer_GapS · 2021-11-21

**Confidence:** 2
**Overall Score:** 7

**Main Review:**


This paper studies the problem of estimating conditional average treatment effects (CATE) with differential privacy guarantees. It proposes a general meta-algorithm that can work with simple, single-stage CATE estimators such as S-learner and more complex multi-stage estimators like DR and R-learner. The authors perform a privacy analysis utilizing the parallel composition property of differential privacy. Finally, experiments demonstrate how differential privacy noise affects the estimated CATE function.

Overall, I think the author considers an interesting problem of addressing privacy concerns in estimating causal effects. Such a problem could be prevalent due to the sensitivity of healthcare data. I appreciate the authors for starting this conversation. However, I do have some questions and concerns, summarized as below.

- This paper focuses on estimating conditional average treatment effect (CATE) while ensuring differential privacy (DP) constraints. This is a bit funny and seems ill-posed. For instance, suppose one wants to compare the CATE of a drug on male and female patients, and the actual CATEs of the male and female population diverge significantly. In this case, ensuring the privacy constraints on gender requires one to add DP noise on the CATE function until one cannot differentiate the CATE on the male and female population. There exists no ideal CATE function in this setting. It makes me wonder if DP formalism is the best tool to address the privacy challenges in causal inference tasks.

- I would suggest writing Sec 2.1 using the formal language of causal inference, e.g., Pearl's structural causal model and the backdoor criterion.

**Summary:**

Review of Paper#67

---

> ### Author Response · Authors · 2021-12-04
> **Reply to Reviewer GapS**
>
> 1. *"Is DP the best tool to address the privacy challenges in causal inference tasks?"*
>
> Differential Privacy has become the standard for protecting privacy in the machine learning and database communities. Differential Privacy focuses on protecting the privacy of individuals in a dataset, not protecting the privacy of groups such as male or female.
> Meeting the DP definition does not require equalizing predictions across distinct groups like male or female.
> As long as the CATE function for gender does not reveal too much information about any one individual, then the algorithm will be able to learn different heterogeneous treatment effects for male and female while satisfying differential privacy.
>
> 2. *"I would suggest writing Sec 2.1 using the formal language of causal inference, e.g., Pearl's structural causal model and the backdoor criterion."*
>
> The paper presents the setup in the potential outcome framework following previous work in the literature (Nie and Wager (2020), Kennedy (2020)).
> We can potentially also present the same setup in a Structural Causal Model (SCM)  framework.
> The two are ultimately equivalent to each other.
> More precisely, the DGP can be represented using the following structural causal model  (Pearl (1995, 2009). Each observed variable $(X,T,Y)$ is a function of parent variables and unobserved background factors $(U_X,U_T, U_Y)$:
> $$X = f_X(U_X)$$
> $$T = f_T(X, U_T)$$
> $$ Y = f_Y(X, T, U_Y)$$
> Our conditional ignorability assumption, $(Y(0), Y(1))\perp T|X$, implies that $U_T\perp  U_Y | X$. By intervening in the SCM we can generate the counterfactual outcome $Y(t)=f_Y(X,t,U_Y)$. The CATE is then
> $\tau(x)=E[Y|do(T=1), X=x] - E[Y|do(T=0), X=x] = E[f_Y(X, 1, U_Y) - f_Y(X, 0, U_Y)|X=x]$.

---

### Official Review · Reviewer_Hjdm · 2021-11-25

**Confidence:** 3
**Overall Score:** 7

**Main Review:**

An estimated CATE function could potentially reveal individual covariates, treatment status, or potential outcomes. The authors propose a general approach for multi-stage learning under differential privacy guarantees and apply it to CATE estimation. They show empirically the trade-off between privacy and estimation accuracy.

The paper is well-written, however familiarity with the differential privacy is assumed by the authors.

The only concern about this manuscript is that it does not have much of theoretical contribution, but on the other hand, it is has great practical value and should be of interest to applied data scientists.

The authors perform a thorough evaluation of the proposed approach. The results show the surprising different behaviors of DP-EBM-S-learner compared to DP-EBM-DR-learner and DP-EBM-R-learner. This shows that choosing a flexible CATE estimator versus a simple estimator is not a trivial decision.

**Summary:**

---

> ### Author Response · Authors · 2021-12-03
> **Thanks ++**
>
> Thanks for the review.  We're glad you like the paper.  We'll try to add more introduction and discussion about DP to the paper for readers who are not familiar with it. Another reviewer also had questions about the theoretical contribution.  We'll modify the paper to make it clearer that the multiple splitting procedure we adopt is the key to being able to prove tight DP bounds for our algorithm -- without it the theoretical guarantees would be much weaker and the impact of DP noise much worse.  Thanks again for the review.

---

### Decision · Program_Chairs · 2022-01-12

**Decision:**

Accept (Oral)

**Comment:**

In this paper, the authors provide a meta-algorithm that uses models with differentiable privacy (DP) guarantees applied to nuisance functions such that a target causal parameter of interest (that uses nuisance functions for estimation) "inherits" those guarantees.

While reviewer opinion was initially split, subsequent discussion addressed most concerns expressed by reviewers. In particular, while the perception of the paper was that it did not have a large theoretical contribution, relative to what is known about DP its contribution in practice was deemed quite important.

As a result, the paper merits publication in CLEAR.